# Accelerating lithium-mediated nitrogen reduction through an integrated palladium membrane hydrogenation reactor

Hossein Bemana ⓘ , Hendrik Schumann, Morgan McKee ⓘ , Senada Nozinovic, Jörg Daniels, Ralf Weisbarth & Nikolay Kornienko ⓘ ✉

Lithium-mediated $N_2$ reduction reaction (LiNRR) is regarded as the most robust route towards electrifying $NH_3$ synthesis. However, in this reaction geometry, hydrogen atoms typically supplied through electrolyte degradation, via $H_2$ oxidation or a combination of both, hampering the efficiency of the process. In this work we provide an alternative H-source by merging a Pd Membrane reactor (PMR) with a LiNRR reactor in a unique dual-reactor setup. Specifically, use a Pd membrane that extracts H atoms directly from $H_2O$ and transfers them across the membrane to an electrodeposited Li layer operating under non-aqueous LiNRR conditions. We show that these $H_2O$-derived H-atoms are used directly to synthesize $NH_3$ in the presence of $N_2$ and electrodeposited Li, thereby opening orthogonal reaction pathways within the metal-mediated nitrogen reduction concept.

The Haber-Bosch synthesis of ammonia from $N_2$ and $H_2$ has supplied the necessary fertilizer to sustain global population growth to the current level of above 8 billion, making its invention as one of the keystones in defining the Anthropocene epoch[1,2]. While its importance cannot be overlooked, its energy use and corresponding greenhouse gas emissions make the search for an alternative ammonia synthetic process a priority in the transition to a sustainable society[3,4]. From a thermodynamic lens, the exothermic nature of ammonia synthesis from gaseous nitrogen and hydrogen ($N_2 + 3H_2 \leftrightarrow 2NH_3 + 46$ kJ.mol$^{-1}$), demands high pressure and low temperature to tilt the equilibrium toward ammonia production. On the other hand, slow reaction kinetics at low temperatures forces Haber-Bosch plants to operate at elevated temperatures, making it an energy-intensive process[5].

There exist various reaction setups proposed to replace the Haber-Bosch process[6–9], among which electrochemical synthesis seems particularly alluring. Here, the reaction is driven by (ideally) renewable electricity under ambient conditions. Among $NH_3$ electrosynthesis systems, the lithium-mediated nitrogen reduction reaction (LiNRR) has been proven to the most reliable method to activate $N_2$ triple bond under ambient conditions to form ammonia[10]. The first mention of LiNRR dates to 1930 by Fichter et al. in which they showed

ammonia is produced by electrolyzing halogenic lithium solution containing alcohols with $N_2$ gas bubbling on platinum cathode[11]. A typical LiNRR system consists of $N_2$ gas, an organic solvent, a lithium salt with high solubility and stability in the solvent, a proton source, and a proton shuttle to deliver the H$^+$ to the cathode.

The community's current understanding of the LiNRR is that the ammonia electrosynthesis initiates with electrodeposition of Li$^+$ on the cathode, followed by decomposition of a non-aqueous electrolyte in contact with metallic lithium and formation of a surface layer called solid-electrolyte interphase (SEI). The SEI is made of a mixture of organic and inorganic compounds, within which the metallic lithium reacts with $N_2$ to form $Li_3N$, then further reacts with a hydrogen source to generate $NH_3$[12–14]. The SEI plays a defining role in ammonia production performance by providing a protective passivation layer for the cathode, and at the same time regulating the reaction by modulating the diffusion of $N_2$, Li$^+$, and H, which considered to be possible bottlenecks within a LiNRR system[15–17]. Unlike traditional electrocatalytic mechanisms which reaction cycle occurs on a two-dimensional surface in a short timeframe, LiNRR takes place in three-dimensional SEI at more extended intervals, adding more complexity to the system[18].

Different strategies have been tested to achieve Faradaic efficiencies (FE) near 100%[19] and ammonia production rates of ~2.5 µmol.cm$^{-2}$.s$^{-1}$[20].

Institute of Inorganic Chemistry, University of Bonn, Bonn, Germany. ✉e-mail: nkornien@uni-bonn.de

These include employing non-aqueous gas diffusion electrodes (GDE)[21], using high $N_2$ gas pressure[19,22], and exploring the use of a variety of proton donors[23–25]. So far, $H^+$ is provided either by sacrificing the proton shuttles and solvent, or by coupling LiNRR with $H_2$ oxidation reaction (HOR)[26], which may pose additional challenges to practical applications. To obtain sustainable HOR, $H_2$ can originate from water electrolysis, which requires separation and then pressurization to prevent the cell from flooding. One might think of using $H_2O$ as the most sustainable of H atoms, but adding $H_2O$ to the LiNRR reactor will result in undesired side reactions, particularly with Li and the SEI. However, $H_2O$ can still be used as an H-source by decoupling $H_2O/H^+$ reduction and hydrogenation steps through the use of a palladium membrane reactor (PMR). The PMR functions through first reducing $H^+$ to *H on a Pd foil (typically 25 μm thick) in an aqueous environment and then transferring *H through the Pd lattice over to an alternative environment on the other side[27]. In such a system, Pd serves as both the cathode for the aqueous chamber and the hydrogen source for the non-aqueous side[28,29]. Here, charge is only used for extracting $H^+$ from an aqueous solution while the hydrogenation reaction occurs through a non-Faradaic process[30,31].

Against this backdrop, we report in this work an approach to lithium-mediated ammonia electrosynthesis, by coupling a LiNRR reaction with a PMR system (PMR-LiNRR). PMR can bypass this by in-situ generation of atomic hydrogen and delivering it to the reaction interface. This is achieved by running two parallel electrochemical reactions, in which the metallic Pd acts simultaneously as the cathode for both electrochemical systems, a separator for the aqueous and organic compartments, and a H-transfer path between the aqueous solution to the LiNRR chamber. As a result, LiNRR hydrogenation follows a reversed geometry, in which H-atoms are relivered from behind the SEI instead of needing to diffuse through it. Further, unlike a conventional PMR, both chambers undergo Faradaic reactions, paving the way to rethinking the hydrogenation reactions. (Fig. 1)

## Results and Discussion

In our initial, proof of concept system, we used tetrahydrofuran (THF) as a prototypical solvent, 1 M lithium tetrafluoroborate (LiBF₄) and small quantities of an alcohol as it was deemed essential towards the generation of a stable SEI that enables $N_2$ and $Li^+$ transport[15,20,32–35]. The control experiments confirmed that ammonia solely originate from LiNRR, including running LiNRR and PMR-LiNRR under Ar instead of $N_2$, and PMR-LiNRR without applying LiNRR current[9]. For the aqueous solution, 1 M $H_2SO_4$ was used for all the experiments and Pt was exclusively used as the anode. We screened the effects of alcohol identity using the PMR-LiNRR reactor, using the LiNRR analogue as a reference. In the latter case, only the alcohol and THF solvent could supply H-atoms while in the PMR-LiNRR the water acts as an alternative H-source in addition. With 0.5% vol. alcohol in the organic compartment and applying −3 mA cm² current to either the Pd in both aqueous and organic sides (PMR-LiNRR) or only to the organic side (LiNRR), we quantified the Li production rate (Fig. 2a). We noted that using ethanol resulted in the highest $NH_3$ yield, as quantified via NMR, similar to most reports in the LiNRR literature[24,36]. In some cases, the LiNRR system did not produce and $NH_3$ and needed the H-atoms supplied via the PMR side. In others, $NH_3$ could be detected without PMR as the small quantity of alcohol was sufficient to produce $NH_3$. It has been observed that increasing the chain length of aliphatic alcohols in LiNRR result in the noticeable drop in FE, mostly assumed that the steric effect of an alcohol affects the diffusion rate of the proton shuttle within the electrolyte and SEI[24,36].

Keeping with ethanol as the alcohol source, we noted that increasing the applied current on the PMR side (while keeping the LiNRR current at −3 mA cm⁻²) boosted the $NH_3$ production rate, providing strong evidence that $H_2O$ was acting directly as the H-source (Fig. 2b). In addition to this, we also noted that, having a continuous current on the LiNRR side was also essential. When keeping the PMR current constant (−3 mA cm⁻²), increasing the LiNRR current also resulted in an increase in $NH_3$ production (Fig. 2c). This illustrates the

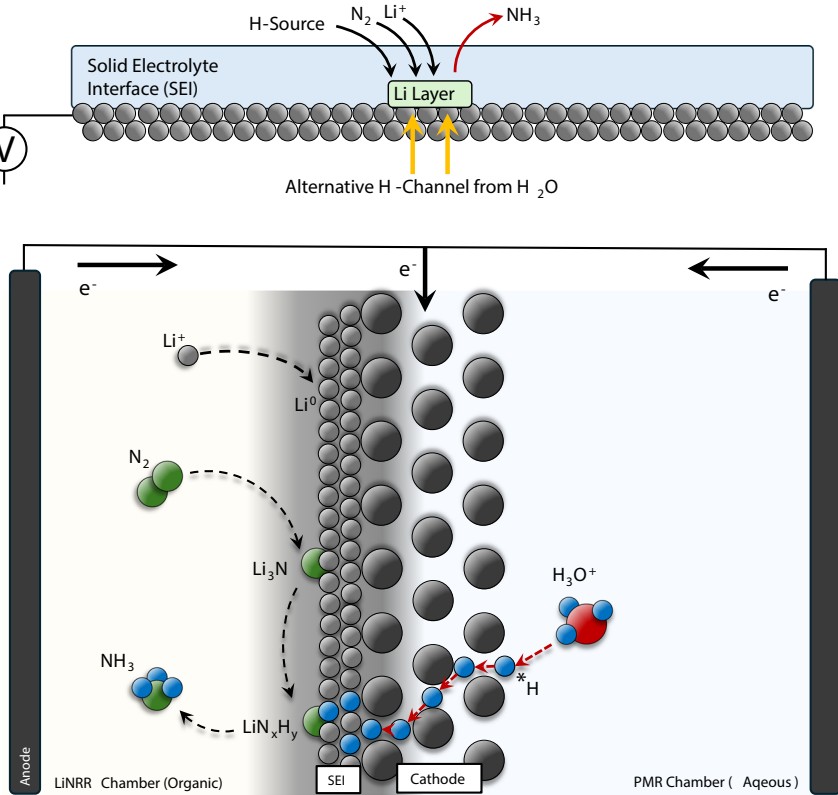

**Fig. 1 | Reactor overview.** Schematic diagram of the palladium membrane reactor coupled with lithium mediated nitrogen reduction reaction.

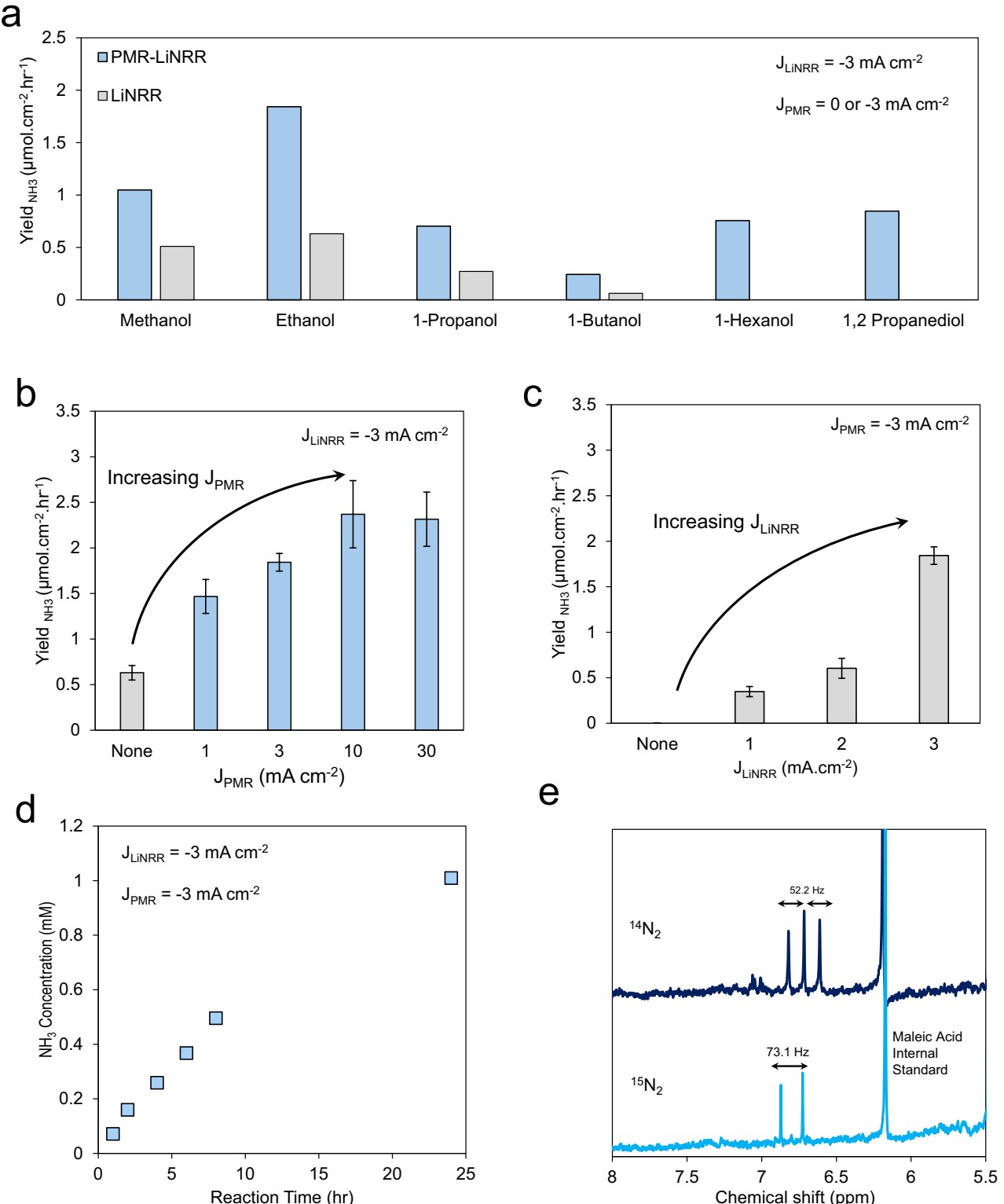

**Fig. 2 | PMR-LiNRR and solo-LiNRR system performance. a** The effects of alcohol identity on the ammonia yield. The application of current to both the aqueous compartment ($J_{PMR}$) (**b**) and organic compartment ($J_{LiNRR}$) (**c**) were essential to maximize $NH_3$ production. The system's performance was stable for the first 24 h, after which the $NH_3$ concentration reached a plateau (**d**). Finally, $NH_3$ production was verified through the use of cleaned $^{15}N_2$, which resulted in a doublet in the HNMR spectrum (**e**). LiNRR electrolyte was 1 M $LiBF_4$ along with 85 mM alcohol dissolved in THF, and PMR electrolyte was 1 M $H_2SO_4$ in $H_2O$. Error bars represent ± standard deviation from three independent measurements. Source data are provided as a Source Data file.

need to continuously reduce $Li^+$ to a metallic Li layer which can react with $N_2$ and the use of the dual-electrochemical system to simultaneously drive the PMR and LiNRR process. We note that we kept the ethanol concentration deliberately low. As the concentration increased beyond 0.5% vol. the ethanol began to increasingly serve as the H-donor and the system behaved more like the standard LiNRR setup, as evidenced by a diminishing rate-enhancement conferred by the PMR part of the reactor (Fig. S7). It has been proposed that high

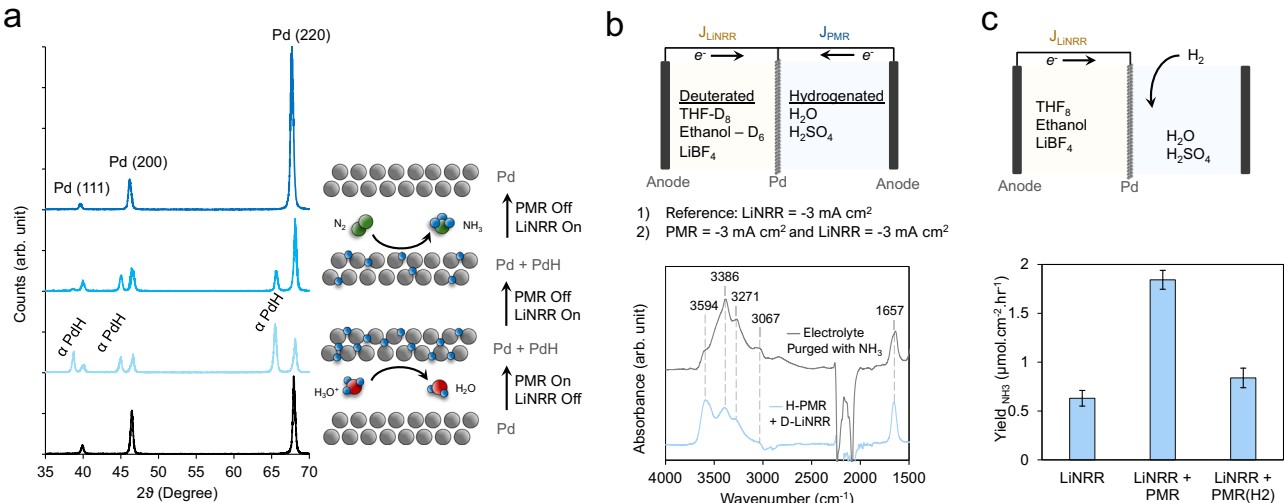

**Fig. 3 | PMR-LiNRR dynamics. a** Dynamics were first investigated through XRD measurements following the application of current in the PMR and LiNRR components. **b** The H-source for NH₃ synthesis was investigated via IR spectroscopy measurements of post-electrolysis solutions when the LiNRR half was fully deuterated. **c** H₂ was shown to be an alternative source of *H species that could hydrogenate the LiNₓ layer, but this route was less effective than direct electrochemical *H generation. Error bars represent ± standard deviation from three independent measurements. Source data are provided as a Source Data file.

ethanol concentration disrupts the N₂/H⁺ balance at SEI, in which ethanol competes with nitrogen to react with metallic lithium, resulting in hydrogen evolution rather than ammonia production[16,32]. Figure S7 shows that incorporating PMR into LiNRR resulted in the negative shift of optimum ethanol concentration. In other words, less ethanol is needed to reach the N₂/H⁺ balance due to the additional hydrogen injected via PMR.

While our typical reaction run lasted 1 h, sufficient to result in a NH₃ concentration that can be reliably measured, we next attempted to probe the longevity of the initial iteration of the PMR-LiNRR setup. When (−3 mA cm⁻²) was applied to both sides of the Pd foil (PMR-LiNRR), the concentration of NH₃ continuously increased within a 24-hr period (Fig. 2d) and plateaued afterwards (Fig. S8), potentially due to reaching equilibrium in the batch reactor, deactivation of Pd foil or deactivation of SEI. Pd foil during PMR sustains physical deformation possibly due to microcracks formed by recombination of hydrogen into H₂[37,38]. To check the functionality of Pd foil, we periodically tested them for C≡N bond cleavage using PMR previously reported by our group[39] and found that the Pd foils get deactivated roughly 50 to 100 h of PMR-LiNRR operation. In addition to Pd membrane failure, SEI deactivation may play a role here. SEI is a complex and dynamic layer in which the structure changes over time as shown in Fig. S6.

Finally, to show that the NH₃ produced resulted from N₂ reduction as opposed to contamination or other sources, we used ¹⁵N₂, scrubbed for impurities, as the reagent and produced ¹⁵NH₃ in comparable quantities (Fig. 2e). While our Faradaic efficiency, quantified here using the total charge passed through both electrochemical reactors, reached only modest levels of up to 5%, this can be readily improved through the use of higher pressures of N₂, advances in reactor engineering and more, as has been accomplished with standard LiNRR systems over the last decade. To understand the reason behind the low FE in our work, the root causes of experimental observations that the N₂ pressure and ethanol concentration affect the LiNRR performance should be pointed out. The LiNRR mechanism consists of different steps, in which the diffusion of Li⁺, H⁺ and N₂ species from bulk solution are the bottleneck of the system. Relative to the slow diffusion steps, the subsequent electrochemical steps are expected to be very fast due to the extreme reducing potential required for Li electrodeposition. Therefore, N₂ and H⁺ concentration directly affect the LiNRR FE, which are proportional to the N₂ pressure and ethanol concentration, respectively. That's why to achieve high FE, high N₂ pressure (>10 bar)

is required, which demands specialized cell design and safety equipment. On the other hand, the present study focuses on an alternative hydrogen transport pathway. Therefore, to simplify the experimental conditions we avoided high N₂ pressure.

Our next endeavor entailed obtaining a mechanistic understanding of the PMR-LiNRR process and providing evidence of the function of the PMR component. To this end, we took to X-Ray diffraction (XRD) to probe the structural dynamics of the Pd layer. We first acquired an XRD pattern of the pre-catalysis XRD foil, which showed the expected reflections of the Pd structure (Fig. 3a, bottom). We next placed the Pd foil in the PMR-LiNRR reactor and turned on the PMR current only (60 min, −3 mA cm⁻²). Taking an XRD pattern immediately after this showed that the foil consisted of both the Pd and αPdH phases, the latter developed from the electrochemical hydrogenation process. Next, the foil was placed back in the PMR-LiNRR reactor and subjected to the LiNRR current only (60 min, −3 mA cm⁻²) while the PMR current was off. The resultant XRD spectrum showed a diminishment of the PdH phase and slight shift to larger 2-theta values as the H atoms partially transferred over to the organic side and the degree of hydrogenation within the Pd decreased. Continuing the LiNRR side for another 120 min at −3 mA cm⁻² resulted in the Pd losing all of its interstitial H-atoms as they presumably transferred over to the organic side of the reactor. Note that the XRD spectra were taken in reflection mode of the portion of Pd facing the aqueous side. The reverse side of the Pd showed mainly signals of the air-exposed SEI film which consisted of a mixture of phases including Pd, PdH₀.₇₅, LiF, LiOH, Li₂(B₁₂H₁₂), and Li₅B₄ (Fig. S5). SEI is a complex and dynamic layer in which the structure changes over time as shown in Fig. S6. Our attempts for in-situ observation of SEI faced technical challenges, primarily due to the difficulty of designing a compact electrochemical cell compatible with our XRD instrument, which required three electrodes and two reaction chambers with separate circulating electrolytes. If the Pd foil is fully covered by a metallic Li layer, the diffusion of H-atoms through lithium becomes an important factor to be considered. However, cryogenic transmission electron microscopy studies revealed that metallic lithium formation is only detectable in the absence of ethanol. Without ethanol, a passivating THF-derived SEI forms on metallic lithium, rendering it inactive for nitrogen reduction. Conversely, the presence of ethanol leads to a poorly passivating SEI with no underlying metallic lithium[33]. A recent study using in-situ neutron reflectometry to monitor SEI formation

revealed that the SEI initially consists of two distinct layers: an inner layer rich in lithium-containing inorganic compounds and an outer organic layer. Under high current density sustained over long durations like in our system, these layers merge and form a thicker and more disordered SEI. This promotes uncontrolled lithium and SEI growth, with inorganic species spreading throughout the structure. As lithiation increases, strains can build up in the inorganic components of the SEI, leading to cracking. These cracks expose fresh lithium to the electrolyte, triggering further decomposition reactions and generating additional inorganic and organic decomposition species. Such cracking could explain the merging of the inner and outer layers into a unified SEI, making them indistinguishable[40]. Based on these, we assumed that H-atoms leaving the Pd foil do not necessarily diffuse through metallic lithium but may instead be directly incorporated into the SEI.

The next demonstration of the PMR-LiNRR system entailed the use of infrared (IR) spectroscopy to elucidate the hydrogen source for $NH_3$ production. To execute this, we modified the PMR-LiNRR setup such that deuterated versions of THF and ethanol were exclusively used in the LiNRR side while the aqueous PMR side was kept to hydrogenated $H_2O/H_2SO_4$ (Fig. 3b). We ran an experiment but with both the LiNRR and PMR currents on (60 min, −3 mA cm$^{-2}$) measured the absorbance of the post-electrolysis solution on the LiNRR side. An increase in the absorbance features of the N-H bands was evident due to the $NH_3$ in the electrolyte that was produced using $H_2O$ as the H-source. The bands matched those of a THF solution that was purged with $NH_3$ gas. In previous reports, it has been shown that a considerable amount of ammonia gets trapped in SEI matrix[16]. Since the hydrogen diffusion path is different in our study compared to the previous reports, it can be assumed that a considerable amount of hydrogen gets trapped at the back side of the SEI facing the Pd foil. Therefore, to qualitatively address the issue, we analyzed SEI by dissolving it into methanol. The relative abundance of N-D over N-H was calculated to be 0.785 (Fig. S14).

As a complementary proof-of-concept we also showed that the $LiN_x$ layer on the Li-NRR side can be hydrogenated without the need for electrochemical steps on the PMR side. To this end, we disconnected the PMR electrochemical cell and instead bubbled $H_2$ through the $H_2SO_4$ electrolyte. In this case, the $H_2$ could spontaneously dissociate into 2H* species and similarly diffuse through the Pd membrane to hydrogenate the $LiN_x$ layer. Using this method, we also recorded a measurable increase in the $NH_3$ production rate, though not as high as when a constant current was applied on the PMR side (Fig. 3C). This is likely due to the higher rate of *H formation from electrochemical $H_3O^+$ reduction at 3 mA/cm$^2$ as compared to $H_2$ dissociated from $H_2$ dissolved in an aqueous electrolyte. Thus, these experiments illustrate the synergy of the PMR-LiNRR system in which Faradaic reactions on both sides are necessary but carry out complementary reactions of $Li^+$ reduction, $H_3O^+$ reduction and subsequent *H transfer. Additionally, we tried an organic proton source in PMR chamber instead of aqueous $H_2SO_4$, which as expected showed improved performance compared to LiNRR (Fig. S15).

In this work, we purposely kept to very standard parameters with minimal modifications to simply demonstrate the concept of enhancing Li-NRR with a PMR reactor. While the performance of this system, in terms of throughput or Faradaic efficiency, does not compete with the state-of-the-art, there is plenty of unexplored avenues to further boost this. For example, increasing the pressure to the 20 Bar commonly used in Li-NRR systems stands to boost reactivity as $N_2$ concentration is increased[15]. Further, exploring nanostructuring/thickness of the Pd or additives like $H_2O$, $O_2$ and other solvent/electrolyte molecules to modulate SEI properties are other readily pursuable follow up endeavors[19,20,41,42].

Looking ahead, while Li is used in the field because of the functional SEI that forms at the electrode surface in LiNRR systems but in theory other metals can also be used[43]. In practice, Ca[44] and Mg[45] were

recently demonstrated as viable alternatives to Li. Other metals like Al have also been hypothesized to be active, though a substantial challenge lies in the formation of an SEI that is sufficiently conductive and porous towards $N_2$, H-atom donors and $NH_3$[43]. We believe that the expansion of the M-NRR + PMR concept may enable the use of such metals because H-transfer becomes decoupled. The PMR-LiNRR has its own intrinsic limitations, notably low $N_2$ dissolution and incompatibility with GDE. The work is in progress to address in this front by modifying the reaction environment and geometry. Another limitation of PMR-LiNRR is the use of palladium as the lithium plating substrate, which can lead to the formation of Pd-Li alloy and degrading the substrate surface[46,47]. Moreover, hydrogen diffusion through palladium can lead to recombination into molecular hydrogen ($H_2$), causing microcrack formation within the palladium foil. This process can deform the material and compromise its long-term structural integrity[37,38].

In all, this work demonstrates that H-atom transfer and metal-mediated NRR can be decoupled through the integration of a PMR reactor with a conventional MNRR setup. Such a unique geometry stands to bring potential benefits to the performance of M-NRR systems as the transport constraints of the SEI layer are relaxed. While this work brought on modest improvements and overall efficiencies still below the state-of-the-art, we anticipate developments in the chemistry and engineering of PMR-MNRR systems may bring this technology closer to practical viability.

## Methods
The list of chemicals used in this study is presented in supplementary information Table S1. A gas-tight H-cell with 30 mL capacity on each side was used as the reaction cell. Palladized palladium foil was placed between two gaskets and fixed between the chambers by a spherical joint clamp. The exposed surface area of palladium to the reaction solutions was -0.785 cm$^2$ on each side. Two platinum electrodes were used as anodes for each chamber. The aqueous side was filled with 15 mL of 1 M $H_2SO_4$, denoted as PMR chamber or aqueous chamber. The non-aqueous side was filled with freshly prepared 15 mL of 1 M $LiBF_4$ dissolved in purified THF with varying amounts of alcohols, denoted as LiNRR chamber or organic chamber. All electrolytes were freshly prepared, and the excess electrolyte was stored in fume hood. A PTFE cap fitted with a rubber O-ring and a screw cap were used to seal the reaction chamber and isolate it from the atmosphere. A glass tube with fritted tip was used to inject gas into LiNRR chamber. The gas flow rate was set to 10 standard cubic centimeters per minute (sccm) by an ALICAT digital mass flow meter. The outlet gas was connected to a gas trap filled with vacuum pump oil. The LiNRR solution was under constant stirring by a magnetic stirrer. All cell parts, separators, and electrodes were sonicated and washed with DI water and acetone and dried in an oven at 110 °C prior to use.

Palladium foils were sequentially sonicated and cleaned with 30% $H_2O_2$, 1 M $H_2SO_4$, then DI water, and dried in 200 °C oven overnight. Electrodeposition of palladium on Pd foils was done in the electrochemical cell described above, filled with 20 mL of 16 mM $PdCl_2$ in 1 M HCl. The potential of the palladium foil was kept at −0.2 V vs Ag/AgCl/KCl$^{sat.}$ reference electrode, until 8 C.cm$^{-2}$ charge passed through the system. The palladized Pd foil was washed with DI water and dried in 110 °C oven[28].

A two-channel BioLogic SP-300 potentiostat was used for electrochemical reactions. The organic and aqueous sides were connected to channel-1 and channel-2, respectively. Both electrochemical systems were performed by two-electrode configuration passing a fixed current (chronopotentiometry) for both aqueous and organic chambers. LiNRR experiments (without PMR) were done by filling the organic chamber and leaving the aqueous chamber empty, connecting only channel-1 of the potentiostat. For the PMR-LiNRR experiments, the Pd foil separating the organic and aqueous chambers was used as the

cathode for both systems, while each of them had their own Pt anode. To control the passage of current on each electrochemical system, Pt anodes were employed as the working electrodes, while Pd cathode was set as the auxiliary electrode. To do so, the counter and reference wirings of both channel-1 and channel-2 were connected to each other, forming a unified auxiliary electrode. The organic chamber was saturated with 15 sccm $N_2$ gas 20 min prior to each experiment and kept purging for entire duration of the experiment. After experimenting with ethanol concentration, $J_{LiNRR}$, and $J_{PMR}$, the optimized reaction condition was set to 1 M $LiBF_4$ with 0.5% v/v ethanol (~85 mM) in THF for organic chamber, 1 M $H_2SO_4$ for aqueous chamber, $J_{LiNRR}$ of 3 mA.cm$^{-2}$, and $J_{PMR}$ of 3 mA.cm$^{-2}$. To test the PMR-LiNRR system with different proton donors, the concentration of each alcohol was set to 85 mM to keep the consistent molar ratio. Potential cycling experiments were performed by looping procedure in EC-Lab program version 11.52 between chronopotentiometry and OCV. Cycling step durations were varied (30 s, 1 min, and 2 min), but the total experiment duration was kept 1 h.

The amount of ammonia in solution was measured by frequency-selective pulsed gradient spin echo nuclear magnetic resonance (NMR)[48], on a Bruker Avance III HD Ascend 500 MHz NMR device. After each experiment, 1 mL reaction solution was added to 50 µL of DMSO-d$_6$ (with 10 mM maleic acid as internal standard) and rigorously stirred. Then, 100 µL of 0.5 M $H_2SO_4$ was added to protonate the ammonia to form ammonium. Using the pulse sequence, the triplet at 6.6–7.0 ppm is assigned to $NH_4^+$, based on the unique splitting pattern associated with the spin 1 $^{14}N$ nucleus and the symmetrical $^1H$ environment. Figure S4 shows the calibration line used in this study, relating the integration of $NH_4^+$ NMR peak to the concentration.

Faradaic efficiency was calculated using Eq. 1, in which F is Faraday constant, C is ammonia concentration measured by NMR, V is electrolyte volume, and Q is the total charge passed. Q for PMR-LiNRR system is the sum of charges passed through each LiNRR and PMR systems (Eq. 2). Note that 3 is the number of electrons transferred to produce each mole of ammonia.

$$FE = \frac{3*F*C*V}{Q} \tag{1}$$

$$Q_{PMR-LiNRR} = Q_{PMR} + Q_{LiNRR} \tag{2}$$

Fourier Transform Infrared Spectroscopy (FTIR) data were collected on PerkinElmer spectrum two FTIR equipped with universal attenuated total reflectance (UATR) accessory and analyzed by SpectrumIR program version 10.7.2. For deuterium-labeled FTIR spectra of PMR-LiNRR is recorded against LiNRR reaction aliquot (without PMR) as the background spectra, to cancel out the common bonding modes, and highlight the additional bonds. Regarding the FTIR spectrum of SEI, the deposited SEI layer was dissolved into methanol immediately after the reaction, and the spectrum is recorded against methanol as the background.

XRD measurements were performed on Siemens Kristalloflex Diffraktometer D5000, with Cu X-ray source, and the spectra were analyzed by Match 4.0 Build 306.

## Data availability

All data will be available upon request to the corresponding author via email for non-commercial purposes. Data will be saved for 10 years and request will be responded to within 10 working days. Source data for the main figures are provided with this paper. Source data are provided with this paper.

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

## Acknowledgements

The authors would like to thank the employees of the Department of Chemistry of University of Bonn, especially Mr. Volker Bendisch, Ms. Simone Weisbarth, Ms. Nicola Friedberger, Mr. Klaus Armbruster, Mr. Norbert Wagner, Ms. Marianne Stanko, Ms. Karin Prochnicki, Ms. Ulrike Weynand, Ms. Hanelore Spitz, and Mr. Thorsten Frings. We acknowledge the University of Bonn and DFG grant KO 7060/4 for funding. This publication was supported by the Open Access Publication Fund of the University of Bonn.

## Author contributions

H.B. led the project, performed experiments, analyzed data. H.S. provided technical support and scientific insight. M.M. provided technical support and scientific insight. S.N. contributed to NMR experiments and analysis. J.D. contributed to XRD experiments and analysis. R.W. contributed to SEM and gave insights into experiments. N.K. supervised the research.

## Funding

## Competing interests

A patent has been filed that partially encompasses this work.
