## [Transparent Peer Review file · Nature Communications]

Driving lithium-mediated nitrogen reduction through a Pd membrane hydrogenation reactor

Corresponding Author: Professor Nikolay Kornienko

Version 0:

Reviewer comments:

Reviewer #1

(Remarks to the Author)

This manuscript proposes the use of Pd membrane to directly use H₂O-derived H-atoms for non-aqueous LiNRR. The topic is interesting. Here are some issues need to be addressed before its further consideration for publication.

1. I'm curious about the experimental device. The authors should provide more details (also including images of equipment).
2. Also, in the schematic diagram in Figure 1, two anodes and one cathode can be seen. So how do the authors couple this two system in the same circuit? Using two different power sources?
3. How is the LiNRR activity if the right part in Figure 1 (H₂O) is replaced by other non-aqueous solvents?
4. How do the authors quantify NH₃? The relevant data and calibration curve should be provided.
5. Strict experiments should be performed to rule out the possible existence of NO₃⁻, NO₂⁻ and N₂H₄ in the electrolyte or Li salt which may also be reduced to NH₃ (ACS Energy Lett. 2019, 4, 9, 2111-2116)
6. The maximum NH₃ yield rate and FE is still unclear in this work. Also, comprehensive surveys should be made to better compare the LiNRR activity with other published works and show the advantage and novelty of this work.
7. Some typos exist in the manuscript; the authors should carefully check it. For example, in Page 6, Fig. 3e is written as Fig. 4e.

Reviewer #2

(Remarks to the Author)

The authors present an innovative approach by developing an integrated palladium membrane hydrogenation reactor to supply hydrogen for lithium-mediated N₂ reduction reactions. This design utilizes H₂O as a sustainable hydrogen source, enabling H-atoms to be delivered from behind the SEI to protonate Li₃N for ammonia generation, as opposed to coupling with the conventional H₂ oxidation reaction. While the concept is novel and holds promise for advancing ammonia synthesis, the conceptual framework and experimental evidence remain underdeveloped. To strengthen the manuscript, I recommend a major revision, particularly in the following areas:

1. The manuscript lacks a discussion on the diffusion performance of H-atoms in metallic lithium. This is a crucial aspect since, after passing through the Pd membrane, the H-atoms must effectively diffuse through a reduced lithium layer before contacting Li₃N, not directly entering the SEI.
2. The authors report that Li-NRR shows no performance when using hexanol or propanediol (Fig. 2a). The reasons behind this observation should be discussed.
3. According to Fig. 2d, the system begins to lose performance after 25 hours. The potential reasons for this instability should be analyzed. Is it due to degradation of the Pd membrane, poisoning of the electrode surface, or structural changes in the SEI? Identifying the root cause would provide insights into the practical feasibility of the proposed system.
4. The authors claim that their strategy enhances Li-NRR with a PMR reactor and demonstrate this using a low ethanol concentration. However, at higher ethanol concentrations, ammonia production primarily results from conventional Li-NRR. This experimental outcome makes it difficult to draw the conclusion that the PMR reactor enhances Li-NRR. The authors should refine their claim and provide additional evidence to validate this enhancement.
5. The reported Faradaic efficiency (5%) is a significant limitation. The authors should analyze the primary reasons for this low efficiency. For instance, is it due to the intrinsic diffusion rate of H-atoms in Pd or Li, or are there other factors contributing to this inefficiency? Identifying the bottlenecks and proposing potential solutions would strengthen the manuscript.
6. Since H transport and its source are central to this study, the authors should go beyond using only ¹⁵N₂ isotopic labeling

to confirm the N source. It is also necessary to use D₂O to trace the H source in ammonia synthesis. This would provide conclusive evidence of H atom origin in their proposed system.

7. The authors repeatedly emphasize that their strategy avoids the challenges associated with conventional Li-NRR by eliminating the HOR reaction. However, the manuscript does not explicitly define what these challenges are. A fair and balanced discussion comparing the proposed strategy with conventional Li-NRR, including its advantages and limitations, is essential to evaluate its true impact.

Reviewer #3

(Remarks to the Author)

This work illustrates a new concept for Li-mediated N₂ reduction to ammonia, where coupling with a Pd membrane reactor enables supply of H from water (the Pd separates the aqueous and non-aqueous/ammonia generation chambers). To my knowledge, I believe that this approach is novel and should be of interest to the broader electrochemistry and ammonia synthesis fields. The authors also provide some mechanistic experiments to illustrate that H₂O-derived H atoms are incorporated into NH₃. The work should be appropriate for Nature Communications. However, there are some important points relating to proper calibration of the data and apparent inconsistencies in the reported results that must first be addressed.

Major comments

1. The results show that ethanol does contribute H to ammonia. It would be good if the authors could provide more in-depth quantification of how much ammonia is generated from ethanol vs. H₂O-derived through the Pd membrane.

2. Fig. 2c, why not increase the LiNRR current above 3 mA/cm², as is done in 2b?

3. The NMR calibration does not appear to cover the appropriate range for the sample data. Only three points are between about 0-1 mM (Fig. S4). As shown in Fig. 2d and S7, the concentration from experiments never gets much above 1, and it's below about 0.5 mM in the first 12 h. Therefore, at least 4 calibration points should ideally be between about 0-1 mM, to get reliable quantification around 0.2-0.5 mM where most of the data fall.

4. Fig S6: For higher ethanol concentrations, why would the ammonia generation rate be lower for PMR-LiNRR than for LiNRR alone? Shouldn't it only be equal or higher, with the only difference being the extra supply of H?

5. P. 5-6, Fig S7 is described as having the ammonia concentration plateau due to breakdown of the catalyst/SEI. However, since the electrolyte is in a batch set up, it seems as though the process will just reach chemical equilibrium at a long time, and that should be the reason for the plateau.

6. According to the reactor results, at least about a third of the ammonia should contain hydrogen from the LiNRR side, even when the PMR side is also operating. However, figure 3B, the FTIR results, suggest that all of the generated ammonia contains hydrogen from the PMR side. It seems as though there should still be some shift in the ammonia peaks due to the approximately one third of the total ammonia that should have been generated from the deuterated solvent on the LiNRR side.

7. The authors should include additional blank controls to ensure accuracy of ammonia detection, such as operating the reactor with no current.

Minor comments

1. There are several errors in the writing, particularly in the abstract, that would be good to improve.

2. P. 2, "demands high pressure and low temperature to tilt the equilibrium toward ammonia production. On the other hand, slow reaction kinetics at low temperatures forces Haber-Bosch plants to operate at elevated temperatures, making it an energy intensive process." This is not exactly an accurate description. The slow reaction kinetics do require elevated temperature, and the thermodynamics dictate high-pressure at this higher temperature that is kinetically required. High pressure is not required at low temperature based on thermodynamics, though.

3. Results 1st paragraph "we quantified the Li production rate" should refer to NH₃ not Li.

4. P.5, Fig. 3c and 3d should be 2c and 2d. There are errors in which figures are referred to in other cases as well, so the authors should proof-read and correct this.

5. Many SI figures are not mentioned in the main text or are mentioned out of order.

Version 1:

Reviewer comments:

Reviewer #1

(Remarks to the Author)

Authors revised their manuscript significantly and replied all of my comments and others. Now I'd like to recommend it for publication in current version.

Reviewer #2

(Remarks to the Author)

the author address most of questions, accept it as it is

Reviewer #3

(Remarks to the Author)

The authors have addressed the comments well and have added the necessary experiments and discussion appropriately; I have no further major comments on this work.

Based on results such as those in figure S7, it seems like the additional H supply from PMR has a negative impact on FE. It would be good if the authors could add some discussion to address this limitation, such as whether the lower FE is unavoidable or if transport or generation rate of hydrogen could be controlled to better match the ratio of H and nitrogen to achieve higher FE using H from the aqueous solvent.

RESPONSE TO REVIEWERS' COMMENTS

Reviewer #1

This manuscript proposes the use of Pd membrane to directly use H₂O-derived H-atoms for non-aqueous LiNRR. The topic is interesting. Here are some issues need to be addressed before its further consideration for publication.

Response: We thank the reviewer for taking the time to evaluate our work and provide constructive comments for further improvements. We have now modified our manuscript in accordance with their suggestions. The details of how we have done so are described below.

1. I'm curious about the experimental device. The authors should provide more details (also including images of equipment).

2. Also, in the schematic diagram in Figure 1, two anodes and one cathode can be seen. So how do the authors couple this two system in the same circuit? Using two different power sources?

Response: To answer the first two questions, have now added a detailed set of information that can be found in the supplementary information. A two-channel bi-potentiostat (BioLogic SP-300) has been used, capable of running two electrochemical reactions at the same time (denoted here as channel 1 and channel 2). The platinum anode of the organic chamber was connected to the working electrode of channel 1, and the platinum anode of the aqueous chamber was the working electrode of channel 2. The counter electrode cables of channels 1 and 2 were connected to each other, then attached to the palladium foil, transforming it into a shared cathode. Both channels were set to floating mode, isolating them from the ground. In this way, the electron flow was controlled by controlling the anodic current rather than the cathodic current.

Page 6 of the supplementary information as well as Figure S1 have been updated.

3. How is the LiNRR activity if the right part in Figure 1 (H₂O) is replaced by other non-aqueous solvents?

Response: We replaced 1M H₂SO₄ with 1M Salicylic acid as a proton donor and 0.1M tetrabutylammonium perchlorate in 1,2-dimethoxyethane solvent. As we had controlled the non-Li side via an applied current which is directly related to the rate of proton reduction to *H on the Pd (and consequently the rate of *H migration to the Li-side of the reactor) we did not expect substantial differences in the ammonia production rate. When keeping reaction parameters consistent (-3 mA.cm⁻² of applied current to each side of the Pd membrane), showing improved ammonia yield rate compared to LiNRR, consistent with our expectations.

This information is now added to the main text, and the data is added to the SI, Fig. S14.

4. How do the authors quantify NH₃? The relevant data and calibration curve should be provided.

Response: The quantification protocol by NMR and the calibration curve have been presented in page 8 of the supplementary information.

5. Strict experiments should be performed to rule out the possible existence of NO₃⁻, NO₂⁻ and N₂H₄ in the electrolyte or Li salt which may also be reduced to NH₃ (ACS Energy Lett. 2019, 4, 9, 2111-2116)

Response: As you correctly pointed out, the nitrogenous contaminants have been a major issue in N₂RR research. That's why we followed "*A rigorous electrochemical ammonia synthesis protocol with quantitative isotope measurements*"¹. We have specifically purified the N₂ and ¹⁵N₂ gasses used as reactants to rule out additional N-based contaminants.

Among all the control experiments, ¹⁵N₂ experiment showed only ¹⁵NH₃, and the LiNRR with Argon gas flow showed no ammonia, ruling out the possibility of false positives due to nitrogenous contaminants in the electrolyte.

6. The maximum NH₃ yield rate and FE is still unclear in this work. Also, comprehensive surveys should be made to better compare the LiNRR activity with other published works and show the advantage and novelty of this work.

Response: The FE data is presented in supplementary information section, Figures S10-S12.

There are two separate current flows in PMR-LiNRR, demanding more charge compared to the solo-LiNRR. Although FE of PMR-LiNRR is higher than solo-LiNRR in certain conditions, the improvement is more evident when comparing the ammonia yield rate.

While the efficiency here is not yet competitive with the best reported values (close to 100% FE) and partial current densities (approx. 1 A/cm²), the conceptual proof of concept can readily be built upon and has the potential to overcome several inherent challenges associated with LiNRR systems.

7. Some typos exist in the manuscript; the authors should carefully check it. For example, in Page 6, Fig. 3e is written as Fig. 4e.

Response: We have now gone through the manuscript and corrected any typos found.

¹ Nature, 2019, 570, 504.

Reviewer #2

The authors present an innovative approach by developing an integrated palladium membrane hydrogenation reactor to supply hydrogen for lithium-mediated N_2 reduction reactions. This design utilizes H_2O as a sustainable hydrogen source, enabling H-atoms to be delivered from behind the SEI to protonate Li_3N for ammonia generation, as opposed to coupling with the conventional H_2 oxidation reaction. While the concept is novel and holds promise for advancing ammonia synthesis, the conceptual framework and experimental evidence remain underdeveloped. To strengthen the manuscript, I recommend a major revision, particularly in the following areas

Response: We appreciate the evaluation from the reviewer and suggestions for improvement. We have now added additional experiments and discussion to address their comments and strengthen our work.

1. The manuscript lacks discussion on the diffusion performance of H-atoms in metallic lithium. This is a crucial aspect since after passing through the Pd membrane, the H-atoms must effectively diffuse through a reduced lithium layer before contacting Li_3N , not directly entering the SEI.

Response: This is a good point brought up by the reviewer. We can imagine two potential scenarios here. 1) The H-atoms must diffuse through a Li layer and 2) If the Li layer is not 100% conformal on the Pd electrode then the H-atoms can directly hydrogenate a LiN_x layer through a $LiN_x/Pd/electrolyte$ interface. However, determining which of these scenarios is dominant requires technical innovation is beyond the scope of this work.

Our attempts for in-situ observation of SEI faced technical challenges, primarily due to the difficulty of designing a compact electrochemical cell compatible with our XRD instrument, which required three electrodes and two reaction chambers with separate circulating electrolytes. If the Pd foil is fully covered by a metallic Li layer, the diffusion of H-atoms through lithium becomes an important factor to be considered.

However, Referring to a paper published by Steinberg et al.², the formation of a metallic lithium layer could be detected only if there is no ethanol in LiNRR system. In this case, a passivating THF-derived SEI forms on top of a metallic lithium, blocking N_2 diffusion. On the other hand, by adding ethanol to the system, poorly passivating ethanol-derived SEI with no metallic lithium sub-layer was observed.

A recent study by Niemann et al. using in-situ neutron reflectometry to monitor SEI formation³ revealed that the SEI initially consists of two distinct layers: an inner layer rich in lithium-containing inorganic compounds and an outer organic layer. Under high current density of $3\text{ mA}\cdot\text{cm}^{-2}$ sustained over long durations like in our system, these layers merge and form a thicker and more disordered SEI. This promotes uncontrolled lithium and SEI growth, with inorganic species spreading throughout the structure. As lithiation increases, strains can build up in the inorganic components of the SEI, leading to cracking. These cracks expose fresh lithium to the electrolyte, triggering further

² Nature Energy, 2023, 8, 138-148

³ JACS, April 2, 2025.

decomposition reactions and generating additional inorganic and organic decomposition species. Such cracking could explain the merging of the inner and outer layers into a unified SEI, making them indistinguishable.

Therefore, it is plausible that the H-atom leaving Pd foil gets directly incorporated into the SEI, without facing a metallic lithium barrier but comprehensively resolving this is left for follow up studies.

2. The authors report that Li-NRR shows no performance when using hexanol or propanediol (Fig. 2a). The reasons behind this observation should be discussed.

Increasing the chain length of aliphatic alcohols in LiNRR has been shown to result in a noticeable drop in FE^{4,5}. Larger alcohols may interfere with the formation of an effective SEI layer. Further, the steric effect can impact the diffusion rate of the proton shuttle within the electrolyte and SEI. Larger alcohol may have lower diffusion rate, limiting the availability of protons at the SEI.

The reasoning has been added to page 5.

3. According to Fig. 2d, the system begins to lose performance after 25 hours. The potential reasons for this instability should be analyzed. Is it due to degradation of the Pd membrane, poisoning of the electrode surface, or structural changes in the SEI? Identifying the root cause would provide insights into the practical feasibility of the proposed system.

Response: Indeed, all the hypotheses mentioned can play a role in performance decline.

Pd foil during PMR sustains physical deformation possibly due to microcracks formed by recombination of hydrogen into H₂, which can block the hydrogen diffusion pathway. To check the functionality of Pd foil, we periodically tested them for a standard reaction, C≡N bond cleavage, using PMR as previously reported by our group⁶ and found that the Pd foils get deactivated roughly 50 to 100 hours of PMR-LiNRR operation.

In addition to Pd membrane failure, SEI deactivation may play a role here. SEI is a complex and dynamic layer in which the structure and particularly the thickness changes over time as shown in Figure S6.

Additionally, the system is in a batch set up, and it is possible that it will reach chemical equilibrium at a long time causing the ammonia concentration to plateau.

4. The authors claim that their strategy enhances Li-NRR with a PMR reactor and demonstrate this using a low ethanol concentration. However, at higher ethanol concentrations, ammonia production primarily results from conventional Li-NRR. This experimental outcome makes it difficult to draw the

⁴ ACS Catalysis, 2022, 12, 5197–5208

⁵ Nature Communications, 2024, 15, 2417

⁶ Chem Catalysis, 2022, 2, 3, 499-507.

conclusion that the PMR reactor enhances Li-NRR. The authors should refine their claim and provide additional evidence to validate this enhancement.

Response: It is true that PMR cannot enhance LiNRR in all possible scenarios. In fact, it can negatively affect the ammonia yield in certain situations.

One way to look at Figure S7 is that the ammonia yield vs. ethanol concentration has an optimum point, which is 0.5% for PMR-LiNRR and 1% for solo-LiNRR. In solo-LiNRR the concentration of ethanol is proportional to the available hydrogen for the reaction, which the incorporation of PMR shifted the optimum ethanol concentration to lower values due to the introduction of a new hydrogen delivery path.

Another way to look at Figure S7 is point by point analysis. At lowest ethanol concentration (0.1%), the ammonia production is minimal in both systems. At 0.25% and 0.5% PMR-LiNRR is clearly producing more ammonia than solo-LiNRR, until at 1% both systems show similar yield. At the highest ethanol concentration (2%), the yield of PMR-LiNRR was lower than solo-LiNRR, which seems counter intuitive. It should be noted that the LiNRR needs a balance between N_2 and H^+ to produce NH_3 effectively. The 2% ethanol concentration is already too high for the solo-LiNRR system, and when it gets integrated with PMR, the N_2/H balance is tipped off even more, resulting in lower yield.

To qualitatively address the issue, we analyzed the N-H and N-D bonding modes of the ammonia in the SEI in Figure S14 in which PMR was filled with deuterated solvents (D_2O and D_2SO_4) but otherwise operating in our optimized conditions. This experiment showed the relative abundance of N-D over N-H is 0.785. It should be kept in mind that in addition to ethanol and PMR, the solvent oxidation (here THF) can also contribute hydrogen to the LiNRR.

5. The reported Faradaic efficiency (5%) is a significant limitation. The authors should analyze the primary reasons for this low efficiency. For instance, is it due to the intrinsic diffusion rate of H-atoms in Pd or Li, or are there other factors contributing to this inefficiency? Identifying the bottlenecks and proposing potential solutions would strengthen the manuscript.

Response: To explain the issue of low FE in our work, the root causes of experimental observations that the N_2 pressure and ethanol concentration affect the LiNRR performance should be pointed out. To do so, we refer to the LiNRR mechanisms proposed by Lazouski et al.⁷ and Andersen et al.⁸. The LiNRR mechanism consists of different steps, in which the diffusion of Li^+ , H^+ and N_2 species from bulk solution are the bottleneck of the system. Relative to the slow diffusion steps, the subsequent electrochemical steps are expected to be very fast due to the extreme reducing potential required to plate Li. Therefore, N_2 and H^+ concentration directly affect the LiNRR FE, which are proportional to the N_2 pressure and ethanol concentration, respectively. That's why to achieve high FE, high N_2 pressure (>10 bar) is required, which demands specialized cell design and safety equipment. On the other hand, the present study focuses on an alternative hydrogen transport pathway. Therefore, to

⁷ Joule, 2019, 3, 4, 1127-1139

⁸ Energy Environmental Science, 2020, 13, 4291

simplify the experimental conditions we avoided using high N_2 pressure. However, this would be a subsequent step to undertake when designing next generation PMR systems.

6. Since H transport and its source are central to this study, the authors should go beyond using only $^{15}N_2$ isotopic labeling to confirm the N source. It is also necessary to use D_2O to trace the H source in ammonia synthesis. This would provide conclusive evidence of H atom origin in their proposed system.

Response: Indeed, this data is in Fig. 3b. The LiNRR compartment was filled with deuterated solvent/ethanol, while the aqueous chamber was kept 1M H_2SO_4 , and the formation of N-H bonds was shown in the FTIR spectrum.

In addition, we analyzed the relative abundance of N-D over N-H in a comparable experiment and this was calculated to be 0.785 (Figure S14).

7. The authors repeatedly emphasize that their strategy avoids the challenges associated with conventional Li-NRR by eliminating the HOR reaction. However, the manuscript does not explicitly define what these challenges are. A fair and balanced discussion comparing the proposed strategy with conventional Li-NRR, including its advantages and limitations, is essential to evaluate its true impact.

Response: The main limitation of HOR is the source of H_2 itself. To obtain sustainable HOR, H_2 can originate from water electrolysis, which requires separation and then pressurization to prevent the cell from flooding. PMR can bypass this by in-situ generation of atomic hydrogen and delivering it to the reaction interface.

The first limitation of PMR-LiNRR is its inability to be used in gas diffusion electrode (GDE) configuration. To circumvent that, a double-PMR cell has been designed to simultaneously pump atomic hydrogen into both anode and cathode. Also, the problem with N_2 dissolution into electrolytes remains, which could be resolved by employing highly N_2 -soluble electrolytes like ionic liquids. The work is in progress. A further limitation of PMR-LiNRR is the use of palladium as the lithium plating substrate, which can lead to the formation of Pd-Li alloy and degrading the substrate surface^{9,10}. Moreover, hydrogen diffusion through palladium can lead to recombination into molecular hydrogen (H_2), causing microcrack formation within the palladium foil. This process can deform the material and compromise its long-term structural integrity^{11,12}.

Appropriate discussions are now added to the text.

⁹ Journal of Applied Electrochemistry, 1995, 25, 48-53.

¹⁰ Journal of Electroanalytical Chemistry, 1989, 210, 445-450.

¹¹ Phys. Chem. Chem. Phys., 2021,23, 13680-13686.

¹² Membranes, 2022, 12, 1132.

Reviewer #3

This work illustrates a new concept for Li-mediated N₂ reduction to ammonia, where coupling with a Pd membrane reactor enables supply of H from water (the Pd separates the aqueous and non-aqueous/ammonia generation chambers). To my knowledge, I believe that this approach is novel and should be of interest to the broader electrochemistry and ammonia synthesis fields. The authors also provide some mechanistic experiments to illustrate that H₂O-derived H atoms are incorporated into NH₃. The work should be appropriate for Nature Communications. However, there are some important points relating to proper calibration of the data and apparent inconsistencies in the reported results that must first be addressed.

Response: We thank the reviewer for their efforts made in finding routes to improve our work. We have addressed their concerns in the detailed response below.

1. The results show that ethanol does contribute H to ammonia. It would be good if the authors could provide more in-depth quantification of how much ammonia is generated from ethanol vs. H₂O-derived through the Pd membrane.

To address this comment, we utilized an isotope labelled aqueous solvent (D₂O + D₂SO₄) and carried LiNRR/PMR out the experiment under standard optimized conditions.

In previous reports, it has been shown that a considerable amount of ammonia gets trapped in SEI matrix¹³. Since the hydrogen diffusion path is different in our study compared to the previous reports, it can be assumed that a considerable amount of hydrogen gets trapped at the back side of the SEI facing the Pd foil. Therefore, to qualitatively address the issue, we analyzed SEI by dissolving it into methanol. The relative abundance of N-D over N-H was calculated to be 0.785 (Figure S14).

To simplify the reaction conditions, we attempted to eliminate ethanol from the system, but no ammonia could be detected if there was no ethanol present. Previous reports propose that in addition to donating protons to LiNRR, ethanol reacts with electrodeposited lithium to form lithium ethoxide, disturbing the SEI matrix which makes the diffusion of N₂, H, and Li⁺ possible. In the absence of ethanol, an impenetrable SEI forms which blocks the diffusion path of reactants^{14,15}. In addition, the limitations of the current NMR quantification protocol for ammonia quantification should be mentioned. As mentioned in the supplementary information, after each reaction 1 mL of aliquot was added to 50 μL of DMSO-d₆ (containing 10 mM of maleic acid as the internal standard) and 100 μL of 0.5 M sulfuric acid in water (transforming ammonia to ammonium ions for detection via NMR)^{16,17}. Regarding the deuterium-labeled reactions in which one chamber was filled with deuterated chemicals, ammonia could exchange ¹H or ²H with the added compounds needed for NMR quantification (DMSO-d₆, maleic acid, sulfuric acid, and water), making precise disambiguation impossible.

¹³ Science, 2023, 379, 707–712.

¹⁴ Joule, 2019, 3, 1127–1139.

¹⁵ Nature Energy, 2023, 8, 138–148.

¹⁶ ACS Energy Lett., 2020, 5, 3, 736–741.

¹⁷ ACS Catal., 2019, 9, 7, 5797–5802.

2. Fig. 2c, why not increase the LiNRR current above 3 mA/cm², as is done in 2b?

Response: This was mainly due to technical reasons. The potentiostat can handle up to 12 V voltage. The total voltage is shown in Supplementary Figure S9 and S10. For a fixed current density of 3 mA.cm⁻², The aqueous side required ~2 V voltage, while the organic chamber showed ~10 V. We could increase the PMR current density (aqueous side) to high values and still it was below 3 V. On the other hand, increasing the LiNRR current density surpassed the 12 V limit, causing the device to overload and turn off the cell.

3. The NMR calibration does not appear to cover the appropriate range for the sample data. Only three points are between about 0-1 mM (Fig. S4). As shown in Fig. 2d and S7, the concentration from experiments never gets much above 1, and it's below about 0.5 mM in the first 12 h. Therefore, at least 4 calibration points should ideally be between about 0-1 mM, to get reliable quantification around 0.2-0.5 mM where most of the data fall.

Response: The calibration line has been updated in page 8 of the supplementary information.

4. Fig S6: For higher ethanol concentrations, why would the ammonia generation rate be lower for PMR-LiNRR than for LiNRR alone? Shouldn't it only be equal or higher, with the only difference being the extra supply of H?

Response: There is an optimal ethanol concentration at which ammonia yields are maximized as reported by other groups^{18,19}, which can be observed in Figure S7 for both solo-LiNRR and PMR-LiNRR systems.

It has been proposed that high ethanol concentration disrupts the N₂/H⁺ balance at SEI, in which ethanol competes with nitrogen to react with metallic lithium, resulting in hydrogen evolution rather than ammonia production.

Figure S7 shows that incorporating PMR into LiNRR resulted in the negative shift of optimum ethanol concentration. In other words, less ethanol is needed to reach the N₂/H⁺ balance due to the additional hydrogen injected via PMR.

5. P. 5-6, Fig S7 is described as having the ammonia concentration plateau due to breakdown of the catalyst/SEI. However, since the electrolyte is in a batch set up, it seems as though the process will just reach chemical equilibrium at a long time, and that should be the reason for the plateau.

Response: We thank the reviewer for this suggestion. Different causes can lead to such observation.

¹⁸ Joule, 2019, 3, 1127–1139

¹⁹ Science, 2023, 379, 707–712

The most straightforward answer would be reaching chemical equilibrium due to the batch setup.

In addition, Pd foil during PMR sustains physical deformation possibly due to microcracks formed by recombination of hydrogen into H₂, which can block the hydrogen diffusion pathway. We periodically tested the functionality of Pd foils and found that they get deactivated roughly 50 to 100 hours of PMR-LiNRR operation.

In addition to Pd membrane failure, SEI deactivation may also play a role. SEI is a complex and dynamic layer in which the structure changes over time as shown in Figure S6. Since in-situ observation of SEI layer was beyond the scope of this study, we just stated the possibilities. The text has been updated on the paper.

6. According to the reactor results, at least about a third of the ammonia should contain hydrogen from the LiNRR side, even when the PMR side is also operating. However, figure 3B, the FTIR results, suggest that all of the generated ammonia contains hydrogen from the PMR side. It seems as though there should still be some shift in the ammonia peaks due to the approximately one third of the total ammonia that should have been generated from the deuterated solvent on the LiNRR side.

Response: We first ran this experiment using a solo-LiNRR experiment run for an equivalent time as the background, which would have the amount of ammonia produced from the conventional route. We then ran a 2nd experiment having both the LiNRR and PMR side operating and took a spectrum of this, so the ND bands here effectively cancelled out and only the NH bands were evident. This was done to highlight the proof-of-concept of this reaction scheme. We have now added experiments and data that show and quantify in relative terms how much ammonia is produced from the PMR and LiNRR routes (Fig. S13).

7. The authors should include additional blank controls to ensure accuracy of ammonia detection, such as operating the reactor with no current.

We followed the rigorous electrochemical ammonia synthesis protocol²⁰, including running the control experiments at least twice, listed below:

1. LiNRR and PMR-LiNRR under Ar instead of N₂, which resulted in no detectable ammonia.
2. PMR-LiNRR without applying LiNRR current, which no ammonia and SEI could be detected.
3. LiNRR without applying PMR current, which the yield was equal to solo-LiNRR.
4. LiNRR and PMR-LiNRR without ethanol, which showed no ammonia.

²⁰ Nature, 2019, 570, 7762, 504-508

RESPONSE TO REVIEWERS' COMMENTS

Reviewer #1 (Remarks to the Author):

Authors revised their manuscript significantly and replied all of my comments and others. Now I'd like to recommend it for publication in current version.

Response: No response needed.

Reviewer #2 (Remarks to the Author):

the author address most of questions, accept it as it is

Response: No response needed.

Reviewer #3 (Remarks to the Author):

The authors have addressed the comments well and have added the necessary experiments and discussion appropriately; I have no further major comments on this work.

Based on results such as those in figure S7, it seems like the additional H supply from PMR has a negative impact on FE. It would be good if the authors could add some discussion to address this limitation, such as whether the lower FE is unavoidable or if transport or generation rate of hydrogen could be controlled to better match the ratio of H and nitrogen to achieve higher FE using H from the aqueous solvent.

Response: We thank the reviewer for their insights. To this end, we have added in a discussion on how next generation systems could sufficiently balance hydrogen transport and reaction rate to augment the overall FE of the system.